# The Glutathione Peroxidase Gene Family in *Nitraria sibirica*: Genome-Wide Identification, Classification, and Gene Expression Analysis under Stress Conditions

**DOI:** 10.3390/genes14040950

**Published:** 2023-04-21

**Authors:** Ziming Lian, Jingbo Zhang, Zhaodong Hao, Liming Zhu, Yuxin Liu, Hao Fang, Ye Lu, Xinle Li, Jisen Shi, Jinhui Chen, Tielong Cheng

**Affiliations:** 1State Key Laboratory of Tree Genetics and Breeding, Co-Innovation Center for Sustainable Forestry in Southern China, Nanjing Forestry University, Nanjing 210037, China; lzming225@163.com (Z.L.);; 2College of Biology and the Environment, Nanjing Forestry University, Nanjing 210037, China; 3Experimental Center of Desert Forestry, Chinese Academy of Forestry, Dengkou 015200, China; nmzhangjb@126.com (J.Z.);

**Keywords:** *Nitraria sibirica*, glutathione peroxidase, salt stress, genomics, gene expression

## Abstract

Plant glutathione peroxidases (GPXs) are the main enzymes in the antioxidant defense system that sustain H_2_O_2_ homeostasis and normalize plant reaction to abiotic stress conditions. However, the genome-wide identification of the *GPX* gene family and its responses to environmental stresses, especially salt stress, in *Nitraria sibirica*, which is a shrub that can survive in saline environments, has not yet been reported. Here, we first report the genome-wide analysis of the *GPX* gene family in *N. sibirica*, leading to a total of seven *NsGPX* genes that are distributed on six of the twelve chromosomes. Phylogenetic analysis showed that *NsGPX* genes were grouped into four major groups (Group I-IV). Three types of cis-acting elements were identified in the *NsGPX* promoters, mainly related to hormones and stress response. The quantitative real-time PCR (qRT-PCR) analysis indicated that *NsGPX1* and *NsGPX3* were significantly up-regulated in stem and leaf, while *NsGPX7* transcriptionally in root in response to salt stress. The current study identified a total seven NsGPX genes in *N. sibirica* via genome-wide analysis, and discovered that NsGPXs may play an important role in response to salt stress. Taken together, our findings provide a basis for further functional studies of *NsGPX* genes, especially in regarding to the resistance to salt stress of this halophyte plant *N. sibirica*, eventually aid in the discovery of new methods to restore overtly saline soil.

## 1. Introduction

Reactive oxygen species (ROS), known as free radicals like O^2−^, OH and non-radicals like H_2_O_2_ and 1O_2_, are originated from incomplete reduction of molecular oxygen in aerobic organisms [1]. Endogenously produced ROS under abiotic stress conditions usually serve as cellular messengers and redox regulators involved in a number of biological processes in plants, but significant accumulation of ROS can cause damage to cellular apparatuses and macromolecules, and eventually program the cells’ death [2]. In many plants, the apoplast, mitochondria, plasma membrane, chloroplast, peroxisomes, endoplasmic reticulum, and cell walls are the primary sites of ROS production [3]. Intracellular ROS are mainly produced at a low level in plants under optimal growth conditions [4]. Previous research revealed that overproduction of ROS produced by abiotic stress conditions in plant cells is extremely reactive and toxic to proteins, lipids, and nucleic acids, which eventually causes cell destruction and death [5]. However, on the other hand, it is also believed that the increased ROS generation in response to stress might work as a signal to activate plant stress response pathways [6]. Therefore, plant cells have evolved sophisticated antioxidant systems to maintain a dynamic balance between ROS production and elimination, thereby minimizing the ROS damage to cells [7]. ROS-scavenging enzymes of plants mainly including glutathione peroxidase (GPX), ascorbate peroxidase (APX), catalase (CAT), superoxide dismutase (SOD), etc. [8]. Among these different ROS-scavenging enzymes, the term glutathione peroxidase (EC 1.11.1.9 for classical glutathione peroxidase and EC 1.11.1.12, phospholipid-hydroperoxide glutathione peroxidase) was introduced by Mills who discovered the reaction with H_2_O_2_ in enzyme preparations from mammalian red blood cells [9]. Compared with mammals GPXs, plant GPXs include cysteine as opposed to selenocysteine in their active site and some of them contain both glutathione peroxidase and thioredoxin peroxidase functions [10]. The first study of plant GPXs is the cloning and characterization of the cDNA from *Nicotiana sylvestris*, which is highly similar to animal GPXs except lacks selenocysteine encoded by the terminal TGA [11]. Plant *GPX*s were discovered to form a distinct cluster by means of a worldwide phylogenetic analysis, indicating that a single ancestral gene may have been the source of all plant GPX genes [12].

The *GPX* gene families have been discovered and analyzed in numerous plants, such as *Citrus sinensis* [13], *Arabidopsis thaliana* [14], *Lycopersicon esculentum* [15] and so on. Several related researches have shown that *GPX* gene help plant to adaptation to various abiotic stresses in multiple ways. In *A. thaliana*, eight members of *GPX* gene family were identified by many researches, four of them were up-regulation under salt stress [16]. Besides, the well-preserved germination rate, seedling growth and chlorophyll content of the *AtGPXL5*-overexpressing seedlings under the stress of 100 mM NaCl indicating the increased salt tolerance of the plant [17]. The up-regulation of *OsGPX2* and *OsGPX4* gene helps several plants to response to drought and oxidative stress, while down-regulated under high salt, heat shock and cold stress in rice (*Oryza sativa* L.) [18]. Under salinity stress, the majority of *CsGPX* genes were down-regulated at particular time points in cucumber (*Cucumis sativus* L.) [19]. Six *ClGPX* genes were identified in watermelon (*Citrullus lanatus*), all of them were obviously up-regulated under salty stress [20]. Different members of eight *TsGPX* genes were coordinately regulated under certain environmental stress circumstances, supporting the critical functions of TsGPXs in salt and drought stress response in *Thellungiella salsuginea* [21]. Research in desert poplar (*Populus euphratica* Oliv.) suggested that the *GPX* gene family genes with induced gene expression pattern belong to a single gene family and are abiotic defend responding to salt stress [22]. Exceptional, in some cases, the expression levels of specific GPX genes may decrease in specific stressed plants, for instance, several GPX genes showed reduced expression in response to drought stress in sorghum (*Sorghum bicolor* L.), which implies that *GPX* genes may be involved in the activation of multiple downstream pathways [23]. In summary, these investigations concluded that plant *GPX*s genes might regulate plant developmental processes, stress reactions, tolerance mechanisms and more [24].

*N. sibirica* is a typical salt-tolerant shrub that thrives in the desert, saline and coastal saline-alkali lands [25], which is a member of the genus *Nitraria Linn.*, and is found in the *Sapindales* clade [26]. Related studies have found that *N. sibirica* belongs to an ecologically adaptabe halophyte with high salt tolerance, which is represented by a certain amount of tissue salt tolerance [27]. These result means *N. sibirica* is a potential pioneer species in high salt environment with strong saline-alkali resistance. In reality, *N. sibirica* is widely distributed in Mongolia, Central Asia, Siberia and China. Due to its high drought resistance and adaptability to saline-alkaline settings, it is the perfect plant for the treatment of salinization and alkalinity [28]. To our current knowledge, the *GPX* gene family has not been investigated in *N. sibirica*, which has influenced the further study on the regulation process of salt stress resistance of *N. sibirica* and restoration overtly saline soil. Thus, being the first, the current study carried out an analysis to identify *GPX* gene family within *N. sibirica*, and revealed the role of *NsGPX* in salt resistance physiology.

## 2. Materials and Methods

### 2.1. Plant Materials and Abiotic Stress Treatment

The seeds utilized for this investigation collected from the Inner Mongolia municipality, Dengkou County, China. The seeds were wrapped in wet sand, kept at 4 °C for more than 60 days to vernalize. The seeds were sown at 23 °C, and they took almost a month to grow to a height of around 25 cm in a 4:1:1 soil mixture of nutrient-rich soil, vermiculite, and perlite. Healthy living *N. sibirica* seedlings of similar growth were given 500 mM NaCl treatment at the same stage of development to mimic salt stress. After 0, 24, and 48 h of salt stress, plant tissues were collected and categorized into root, stem, and leaf sections for subsequent experiments. Following that, every single one of the plant tissues were gathered, quickly frozen with liquid nitrogen, and kept at −80 °C immediately to extract RNA [28]. The dates of sample collection were 15 August to 17 August 2022.

### 2.2. Identification of GPX Genes in N. sibirica

The HMM (Hidden Markov Model) file of the GPX domain (Pfam: PF00255) from the Pfam database (http://pfam.xfam.org/, accessed on 10 August 2022), using a local software, HMMER3.1 [29], to make comparison in the unpublished genome database of *N. sibirica*. As for BLASTP, we used BLASTP software (v2.90) eight *A. thaliana GPX*s amino acid sequences as an enquiry by e-value set to 1 × 10^−5^. The amino acid sequences of eight *AtGPX*s were retrieved from the TAIR *Arabidopsis* genome database (http://www.arabidopsis.org/, accessed on 10 August 2022) [30]. Seven *NsGPX* genes were recognized by merging the result of these two methods and named after *AtGPX*s. In order to confirm the correctness, we used SMART (http://smart.embl-heidelberg.de/, accessed on 10 August 2022) and the online CD-search tool (https://www.ncbi.nlm.nih.gov/Structure/bwrpsb/bwrpsb.cgi, accessed on 10 August 2022).

### 2.3. Characterization of GPX Genes Family in N. sibirica

Using online ProtParam tool (http://web.expasy.org/protparam/, accessed on 10 August 2022) to examine the physico-chemical characteristics including MWs (molecular weights) and isoelectric points of NsGPXs. The subcellular localization of NsGPX proteins was prophesied from the DeepLoc-2.0 tool (https://services.healthtech.dtu.dk/services/DeepLoc-2.0/, accessed on 10 August 2022) [31] (Appendix A). The gene structures of *NsGPX*s were created via TBtools software (V 1.098; https://github.com/CJ-Chen/TBtools, accessed on 10 August 2022) [32]. by Employing the MEME website (https://meme-suite.org/meme/db/motifs, accessed on 10 August 2022) [33], the conserved motifs of NsGPX protein sequences were recognized. The Plant CARE online tool (http://bioinformatics.psb.ugent.be/webtools/plantcare/html/, accessed on 10 August 2022) was used to examine the *cis*-acting elements within the *NsGPX* gene promoters, and the figure was produced by TBtools (V 1.098).

### 2.4. Phylogenetic Tree Analysis of GPX Proteins in N. sibirica

To clarify the evolutionary relationship of the *NsGPX* gene family, we generated a phylogenetic tree for *N. sibirica*, *A. thaliana*, *Vitis vinifera* and *O. sativa* protein sequences. ClustalW served as the tool for sequence alignment, pruning of the sequence was accomplished using trimAL [34], iqtree2 (Bootstrap = 1000) was used to construct evolutionary trees, which were beautified using ggtree2 package in R language.

### 2.5. Expression Analysis of GPX Genes in N. sibirica

The Eastep^®^ Super Total RNA Extraction Kit (Promega, Shanghai, China) was utilized to extract RNA of experimental seedlings; then the RNA was reverse transcribed into single-stranded cDNA using a reverse-transcription kit (Vazyme, Nanjing, China), which containing DNA clearance, after RNA quality control by gel electrophoresis. Using a LightCycler 480II (Roche, Basel, Switzerland), quantitative real-time PCR (qRT-PCR) was performed with the AceQ qPCR SYBR Green Master Mix (Vazyme, Nanjing, China). In order to detect expression of *NsGPX*s under salt stress, particular qRT-PCR primers were designed in NCBI (https://www.ncbi.nlm.nih.gov/tools/primer-blast/index.cgi?LINK_LOC=BlastHome, accessed on 1 May 2022) (Appendix A). As for reference genes, we chose *NsHis* and *NsAct*, which maintained stable expression under salt stress [35]. All of the primers were verified the specificity and the optimal annealing temperature by gel electrophoresis. All the primers were specific and the optimal annealing temperature was 57 °C (Appendix A). To further examine the expression responses of the *NsGPX* gene family under salt stress since *N. sibirica* is a well-known halophyte shrub, we treated *N. sibirica* seedling with 500 mM NaCl solution and collected different tissues for expression quantification of random chosen genes, and the data were normalized based against two reference genes. All the primers used were specific (Appendix A).

## 3. Results

### 3.1. Identification and Characterization of NsGPX Genes

By merging the result of HMM search and BLASTP, we identified 7 *GPX* genes in *N. sibirica*, and each individual was given a name based on its *Arabidopsis* homolog. (Table 1). NsGPX proteins ranged from 144 aa (NsGPX6) to 480 aa (NsGPX3, NsGPX4) in length and their molecular weights (MWs) ranged from 16.06 kDa to 54.09 kDa, while their pI ranged from 5.34 to 9.98, with an average of 7.95. The online prediction of subcellular localization showed that two NsGPX proteins, including NsGPX3 and NsGPX4, were located in mitochondrion or plastid, while NsGPX5 was mostly located in plastid. Others are distributed in other part of the cell: cytoplasm (NsGPX7), nucleus (NsGPX6), extracellular (NsGPX2) and endoplasmic reticulum (NsGPX1) (Appendix A).

### 3.2. Phylogenetic Relationships of NsGPX Genes

In order to reveal the evolutionary history of *NsGPX* genes, the predicted GPX protein sequences from *N. sibirica*, *A. thaliana*, *V. vinifera* and *O. sativa* were aligned and then were used to reconstruct the phylogenetic tree by the neighbor-joining (NJ) method. The results of the research demonstrated that there were four major combinations among all GPXs, i.e., Group Ⅰ–Ⅳ (Figure 1).

In general, GPXs clustering into the same group may possess comparable functions. NsGPXs and GPX proteins from other species were spread in each group. Especially, NsGPX1 and NsGPX6 were clustered into Group Ⅳ and Ⅱ, respectively, while NsGPX3/4 and NsGPX2/5/7 were clustered into Group Ⅰ and Ⅲ. Furthermore, the expansion of NsGPX3/4 and NsGPX2/5/7 were both lineage-specific, implying a particular function which might be related to biological characteristics of *N. sibirica*.

By mapping *NsGPX* genes into chromosomes, we found that these seven *NsGPX*s were distributed on six of the twelve chromosomes. Most of them corresponded to one chromosome, except for *NsGPX3* and *NsGPX4* on Chr4. (Figure 2).

### 3.3. Collinearity Analysis of NsGPX Genes

Intraspecies collinearity analysis showed that no gene pair existed among these *NsGPX*s. Interspecific covariance analysis demonstrated substantial orthologs of the *GPX* genes when it comes to *N. sibirica*, *A. thaliana* and *V. vinifera* (Figure 3). 5 *NsGPX* genes exhibited syntenic links with 4 *AtGPX*s and 4 *VvGPX*s. Predominantly, most homologues of *A. thaliana* and *V. vinifera* have colinear affiliations with *NsGPX*s, which implies that whole-genome duplication (segmental duplication) events participated in a significant role in *NsGPX* gene family evolution in the *N. sibirica* genome. Notably, *NsGPX3* and *NsGPX4* link with one gene both of *A. thaliana* and *V. vinifera*, which means these genes share closer relation.

### 3.4. Evaluation of Gene Structures and Conserved Motifs of NsGPX Genes

In order to fully comprehend how the genes in the NsGPX family have expanded, the gene structure (exon and intron arrangement) of NsGPXs was scrutinized. The assessment outcome indicated that the number of exon in each gene varied from 4 (*NsGPX6*) to 12 (*NsGPX3/4*). No UTR was predicted for *NsGPX2*, *6* and *7*, while the rest *NsGPX* genes possessed 5’ and 3’ UTR.

Conserved domain analysis showed that all seven *NsGPX* genes contained Thioredoxin_like superfamily (Motif1) and most of them possessed Motif2 and Motif10 except for NsGPX7. The identical motif type and order between NsGPX3 and NsGPX4 implied similar protein functions of this gene pair (Figure 4). Our analysis concluded that, unexpectedly, GPX family members within a group did not share the exact similar genetic architecture and evolutionary relatives.

### 3.5. Expression Patterns of NsGPXs in Response to Salt Stress

The prediction of *cis*-acting elements revealed that three main types of *cis*-acting elements, i.e., those related to plant hormone signaling, environmental stress responses, and MYB-binding sites, were enriched in the promoter regions of *NsGPX*s. The evaluation of *cis*-regulatory elements in the *NsGPX*s displays that five phytohormone-correlated comprising auxin, abscisic acid (ABA), gibberellin (GA), methyl jasmonate (MeJA), and salicylic acid (SA), responsive elements comprising MYB binding sites, MBS, etc. (Appendix A). All the *NsGPX* genes contain MYB binding sites in their promoter regions, and most of them possessed *cis*-elements related to more than one phytohormone, indicating that the *NsGPX* gene family was extensively involved in plant growth and development, as well as in response to environmental stresses like drought and salinity (Figure 5).

qRT-PCR analysis demonstrated that most members of *NsGPX* gene family transcriptionally responded to salt stress both in root and shoot from 24 h after salt stress until 48 h.

Notably, the specific salt stress responsive NsGPX genes varied between different tissues. Specifically, *NsGPX7* was significantly upregulated 24 h after exposed to salt stress and kept stable at 48 h in root (Figure 6a). In shoot, *NsGPX1* and *NsGPX3*, instead of *NsGPX7*, were transcriptionally up-regulated in response to salt stress (Figure 6b,c). Interestingly, the transcriptional response of *NsGPX1* obviously lagged behind *NsGPX3* both in stem and leaf. The expression of *NsGPX1* was still kept in low level at 24 h but increased a lot at 48 h after exposed to salt stress, while *NsGPX3* already transcriptionally up-regulated at 24 h and peaked at 48 h. In particular, the transcriptional response of *NsGPX4* was not similar to *NsGPX3* under salt stress, which means these two genes have different expression patterns under salt stress.

## 4. Discussion

ROS are commonly produced in plants under stress, which plays a substantial part in plant physiology. In order to prevent excessive ROS from affecting the normal growth of plants, plants have evolved the ability to neutralize excessive ROS, among which GPXs are considered to be the greatest possible ROS scavenger [36]. In *Ammopiptanthus nanus*, research exhibited that GPX family members might participate in the stress response by contributing to reactive oxygen species scavenging [37]. Evidence from diversified studies on GPXs from different plants further demonstrated their crucial physiological and developmental roles. By way of example, *AtGPX* isoenzymes may function to detoxify H_2_O_2_ and organic hydroperoxides using thioredoxin in vivo and may also be involved in regulation of the cellular redox homeostasis by maintaining the thiol/disulfide or NADPH/NADP balance under salt stress [38]. Moreover, every single one with the exception of *PpaGPX1* of *PpaGPX* genes presented up-regulated expression after the highest level of respiratory climacteric in the late stage of ripening, and under heat + 1-MCP treatment, which means *PpaGPX*s had a more significant regulating function in the later stage of peach fruit ripening [39].

With the goal to understand the response pattern under salt stress of *N. sibirica*, considered the important role of GPX, genome-wide analyses were performed in *N. sibirica* to identify *GPX* gene family members and their expression patterns in response to salt stress. To the best of our current knowledge, the *GPX* gene family, which responds to salt stress conditions, has not yet been reported in *N. sibirica*. And we found 7 *GPX* genes across 6 chromosomes, which can be divided into 4 groups (Figure 1). Previous studies have found that *GPX* gene family got different pattern of sub-cellular localization [40], while our study showed that *NsGPX* got similar pattern, just like *Populus trichocarpa GPX*s [41], this should be further investigated by subcellular localization in the future. Fluctuation in the GPXs gene numbers between distinct species of plants could potentially be attributed to gene replication experiences, study on the *Brassica napus GPX* gene family revealed that segmental duplication, purifying selection, and positive selection burden did exist during the evolution of *BnGPX*s [24]. Interspecific covariance analysis uncovered whole-genome duplication events performed a crucial part in *NsGPX* gene family evolution in the genome. Notably, *NsGPX3* and *NsGPX4* showed similar location on chromosomal and collinearity, they may be tandem repeat sequences (Figure 2).

The number of motifs varied considerably in the *NsGPX* gene sequences, ranged from 2 to 11, and the number of CDSs ranged from 5 to 12. Additionally, the discrepancy in the number of introns in the gene structure of GPXs genes implies that some variation exists in this given gene families, which proves that members of the *NsGPX* gene family are functionally manifold. Remarkably, *NsGPX3* and *NsGPX4* showed parallel exon-intron organization and conserved motifs, which indicates that both of these genes might play a component in the comparable roles pertaining to multiple abiotic cues (Figure 4). Additionally, similar breakthroughs have been reported on *Gossypium hirsutum* [42] and *Triticum aestivum* [43]. Analysis of the cis-acting elements showed that all the *NsGPX* genes contain MYB binding sites, 4 of *NsGPX*s contain MYB bonding site associated with drought-inducibility and light responsive element. Most *NsGPX*s contain the cis-acting elements related to plant hormones such as ABA, MeJA, GA, SA, and auxin, meaning they may portray roles in response to salt stress (Figure 5).

According to the expression profiling of *GPX* genes, number of abiotic and biotic stresses have been revealed to induce the GPX genes. The qRT-PCR result shows that 5 *NsGPX*s have increasing levels of expression in *N. sibirica* leaf, stem and root from 24 h after salt stress until 48 h, which means *NsGPX*s may portray an important role in response to salt stress, especially in scavenging excessive ROS (Figure 6). This result is also of considerable significance to further understand *N. sibirica*’s mechanism of salt-tolerance. Still, the expression levels of *NsGPX*s in various tissues are significantly divergent, which is similar to the study on the *Lotus japonicus* GPX gene family [44]. The high expression of *NsGPX1* in all plant parts under salt stress indicated that *NsGPX1* was involved in the main process of salt stress resistance, while the significantly improved expression of *NsGPX3* in leave after salt stress showed that its function may be similar to *AtGPX3*, which regulates REDOX status through H_2_O_2_, interacts with phospholipase ABI1/ABI2, activates plasma membrane Ca^2+^ and K^+^ channels, closes stomata, and participates in ABA and stress regulatory signal pathways [45]. In addition, though *NtGPX3* and *NtGPX4* are very similar in sequence composition and domain, which means they may have similar protein functions, their expression patterns are completely different under salt stress, indicating that the differentiation of their expression patterns can also affect the function of genes. Besides, we found no intraspecies collinearity in *NsGPX* gene family and no tandem repeat sequences through mcscanx, which means *NtGPX3* and *NtGPX4* are not duplicating genes (Appendix A). This result differentiates from GPX gene family in cucumber, which the segmental along with tandem duplication has been revealed as the reliable source for the GPX gene family expansion [19]. Research conducted on Soybeans identified that *GPX* genes displayed substantial differential expression of such genes in response to oxidative, drought and salinity stresses in plant root tissue [46]. When *N. sibirica* roots under salt stress, the high expression of *NsGPX7* revealed that its function in salt stress resistance may focus on the scavenging of excess ROS in roots. Related studies in *A. thaliana* have showed that both *AtGPX1* and *AtGPX7* are related to the reduction of H_2_O_2_ under photooxidative stress, the functions of both genes are not completely redundant [47], which means depletion of *GPX1* and *GPX7* expression may affect the reduced tolerance to abiotic stress. Still, further research is needed to determine the molecular mechanism of *NsGPX*s function and to fully understand how exactly *NsGPX* gene family contributes to salt stress response by using various functional validation analyses, such as overexpression, knockout via CRISPR/Cas system, etc.

## 5. Conclusions

The current study identified a total seven *NsGPX* genes in *N. sibirica* via genome-wide analysis. For purpose of in-depth knowledge into the evolution of *NsGPX* gene family, gene structure, phylogenetic, conserved motifs, cis-elements, and expression profiling against salt stress treatments were accomplished. *NsGPX* genes expressed differentially in different tissues under salt stress, most of them have increasing levels of expression from 24 h after salt stress until 48 h, which means *NsGPX*s may play an important role in response to salt stress. These discoveries may lay a foundation for further research of *NsGPX*s in *N. sibirica* developmental processes and response to variable abiotic stresses, then help to further understand the salt resistance of *N. sibirica*, eventually aid in the discovery of new methods to restore overtly saline soil.

## 6. Summary

The current study filled the gap in related studies through the analysis of *NsGPX* gene family, and preliminarily explored the role of *NsGPX* in salt stress response, providing the basis for in-depth exploration stress response mechanism of *N. sibirica*. Our study is also fundamental to functional validation analyses, which is necessary to fully understand how exactly *NsGPX* gene family contributes to salt stress response. All these further studies will be the furtherance of utilization and ecological restoration of saline soil.

## Figures and Tables

**Figure 1 genes-14-00950-f001:**
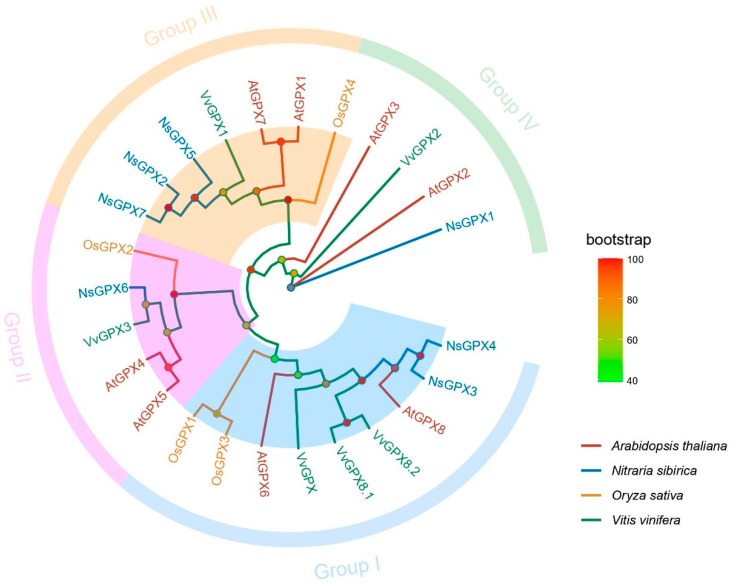
A neighbor-joining Phylogenetic analyses of 25 plant glutathione peroxidase (GPX) proteins. The plant GPXs include 8 AtGPXs (red), 5 OsGPXs (yellow), 6VvGPXs (green) and 7NsGPXs (blue), symbolized by unique colors.

**Figure 2 genes-14-00950-f002:**
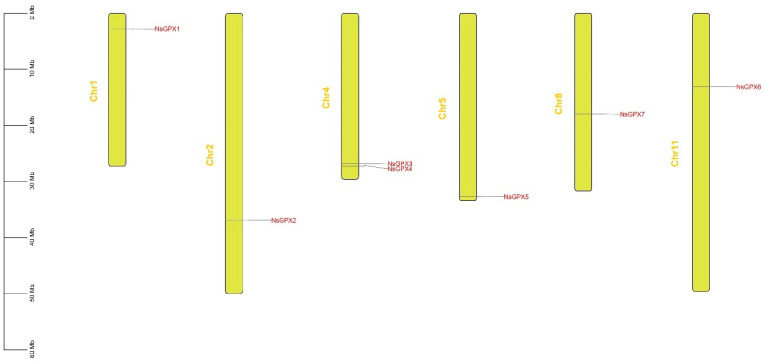
Chromosome location of *NsGPX* genes. Each chromosome gene names of each *GPX* genes on the right side coincide to their approximate locations.

**Figure 3 genes-14-00950-f003:**
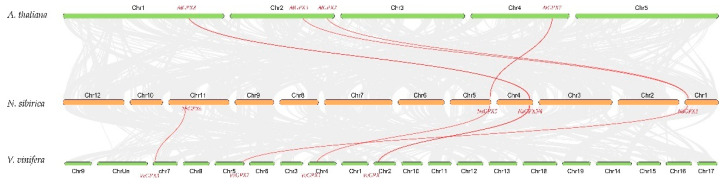
Synteny analyses of *GPX* genes in *N. sibirica*, *A. thaliana*, and *V. vinifera* chromosomes. Background gray lines represent the collinear blocks within the genomes of *N. sibirica* and other plants, meanwhile the red lines highlight the syntenic *GPX* gene pairs.

**Figure 4 genes-14-00950-f004:**
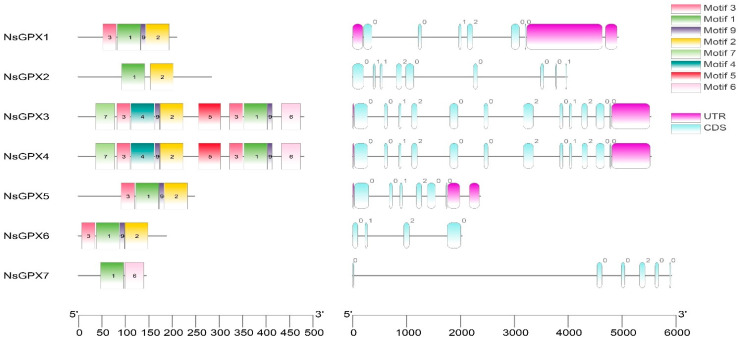
Domain distribution and conservative motif distribution of *NsGPX* genes. The names of *NsGPX*s conserved motifs configurations are given. Specific color blocks represent unique motifs. The CDS regions of *NsGPX*s are indicated by light blue, whilist the *NsGPX*s UTR regions are represented by purple. The numbers in the right of color blocks are phase number of CDS.

**Figure 5 genes-14-00950-f005:**
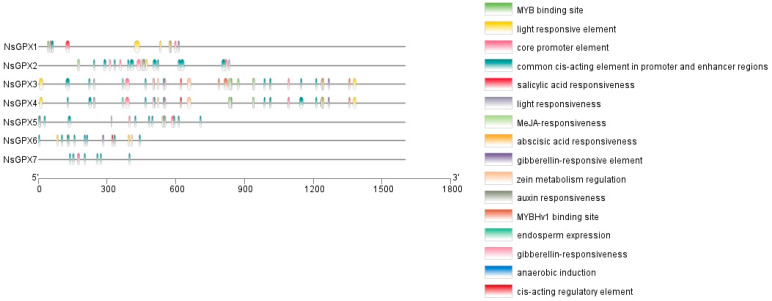
Evaluation of *cis*-regulatory elements in the *NsGPX*s promoters’ regions that are linked with assorted hormone- and stress-responsive elements. Specific color blocks represent unique identified *cis*-elements.

**Figure 6 genes-14-00950-f006:**
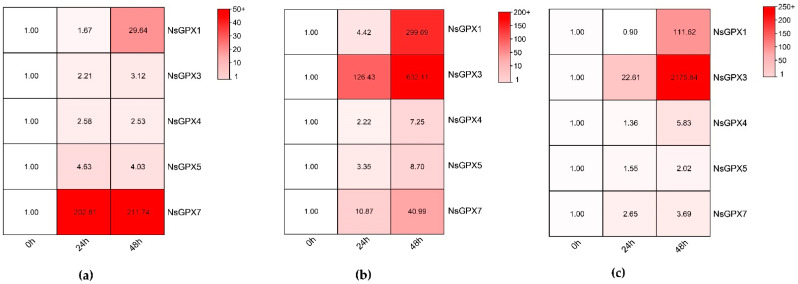
Expression profiles of *NsGPX*s under abiotic stress in root (**a**), stem (**b**), leave (**c**).

**Table 1 genes-14-00950-t001:** The seven members of the *NsGPX* gene family identified from the *N. sibirica* genome.

Gene	Gene ID	Genomic Position	Gene Length (bp)	CDS Length (bp)	Protein Length (AA)	MW (kDa)	PI	Subcellular Localization
*NsGPX1*	NISI01G0182.1	Chr1	2246	624	209	23.76	6.18	Endoplasmic reticulum
*NsGPX2*	NISI02G2276.1	Chr2	843	843	283	32.37	9.98	Extracellular
*NsGPX3*	NISI04G1554.1	Chr4	2186	1000	480	54.06	8.62	Mitochondrion|Plastid
*NsGPX4*	NISI04G1598.1	Chr4	2191	1000	480	54.09	8.73	Mitochondrion|Plastid
*NsGPX5*	NISI05G1741.1	Chr5	1211	738	247	27.25	9.18	Plastid
*NsGPX6*	NISI11G1011.1	Chr11	560	560	144	21.07	7.61	Nucleus
*NsGPX7*	NISI08G0581.1	Chr8	429	429	187	16.06	5.34	Cytoplasm

## Data Availability

The data supporting the findings of this work are available in the manuscript and its Appendix A.

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
