# Peer review of "The Glutathione Peroxidase Gene Family in Nitraria sibirica: Genome-Wide Identification, Classification, and Gene Expression Analysis under Stress Conditions"

_genes, 2023, doi:10.3390/genes14040950_

Round 1

Reviewer 1 Report

The manuscript is well written, but some corrections are needed. Please improve the introduction and methodology part.

Author Response

Point 1: The manuscript is well written, but some corrections are needed. Please improve the introduction and methodology part.

Response 1: Thank you for the comment. We rephrased the introduction and methodology part in a more logical way in the revised manuscript, mastakes in grammer and spelling have been corrected. Here is an example from the methodology part:

Line 145-162: RNA was extracted using the Eastep® Super Total RNA Extraction Kit (Promega, Shanghai, China); the RNA was then reverse transcribed into single-stranded cDNA using a reverse-transcription kit (Vazyme, Nanjing, China), which containing DNA clearance, after RNA quality control by gel electrophoresis. Quantitative real-time PCR (qRT-PCR) was performed using a LightCycler 480II (Roche, Basel, Switzerland) with the AceQ qPCR SYBR Green Master Mix (Vazyme, Nanjing, China). Specific qRT-PCR primers were designed in NCBI ( https://www.ncbi.nlm.nih.gov/tools/primer-blast/index.cgi?LINK_LOC=BlastHome, accessed on 1 May 2022) to detect expression of NsGPXs under salt stress (Table S2). As for reference genes, we chose NsHis and NsAct, which maintained stable expression under salt stress[35]. All of the primers were verified the specificity and the optimal annealing temperature by gel electrophoresis. All the primers were specific and the op-timal annealing temperature was 57°C (Figure S1). To further examine the expression responses of the NsGPX gene family under salt stress since N. sibirica is a well-known halophyte shrub, we treated N. sibirica seedling with 500 mM NaCl solution and col-lected different tissues for expression quantification of random chosen genes, and the data were normalized based against two reference genes. All the primers used were specific (Figure S2).

Reviewer 2 Report

In this manuscript, a genome-wide analysis in N. sibirica revealed 7 GPX genes distributed in 4 different groups. 3 of these genes were upregulated regarding different tissues as leaves, stems and roots. In general, there are some concerns that need to be solved.

 Methods

-With regard to RNA quality used in the experiments.

Was the total RNA treated with DNAse to remove DNA contamination?

If yes, what was done to confirm the DNAse efficacy used to removal DNA?  Was the RNA tested in + compared to –RT reactions?;

No information on these points were given and they are indispensable to ensure that the results truly represent transcripts.

-The authors not explain how they proceeded to normalize the qPCR data”. No reference gene with stable expression under salt stress was informed…  This is crucial to ensure a precise transcript quantification.

- About the primers. Have the authors obtained the annealing temperatures experimentally? Primer pair efficiencies were performed?

- Also, It is also very important to verify the primer specificity experimentally. On this, were melting curve performed? Or were PCR products visualized in a agarose gel?

-The authors used data from an unpublished genome, however, I think that would be important to have the sequences of NsGPX genes and (or cDNAs) available to the readers.

Results

-Avoid including methodology information in result section. It occurred several times!

- I would like to see the protein subcellular localization using the DeepLoc-2.0 tool https://services.healthtech.dtu.dk/services/DeepLoc-2.0/. It uses machine learn and could give a more precise localization.

-Regarding the Figure 1, I don’t understood the classification of NsGPX proteins. In fact, regarding Arabidopsis proteins AtGPX, they was not well named in some cases since AtGPX1 and 7 are close related, for example, perhaps a good classification would be to name AtGPX1 as 1a and AtGPX7 as 1b… Thus, this group could to be named as group I, with NsGPX proteins (2, 5 and 7) named for NsGPX1a, 1b and 1c… and successively…

-Why the authors not mentioned in the results the expression data of NsGPX2 and 6 genes? In addition, for me, it is not clear that the expression data presented in Figure 6 were normalized against any reference gene. This needs to be clarified.

- Discussion: No table or figures were informed when discussing the results.

- The manuscript needs to be carefully checked regarding grammar and spelling.

Minor points:

Revise lines 12-15 (abstract). The link and construction of sentences need to be improved.

Line 30. OH-

Line 108. Please, includes the year. “The dates of sample collection were 15 August to 17 August. “

Lines 110 to 119, revise.

Line 124. Use NsGPXs (regular form; non italic) when referring to proteins.

Line 126. “gene” instead “genes”

Line 135. thaliana instead Thaliana

Line 145. The authors state that specific qRT-PCR primers were designed in NCBI. What software they used?... the link http://bioinformatics.psb.ugent.be/webtools/plantcare/html/ have no sense with primer design.

Reviewer 3 Report

Following are few suggestion towards improve the standard of the manuscript.

Comments to the Authors

1.      The introduction is too general, it should be improved.

2.      Authors citation of the plant should be given.

3.      In materials and methods: Authors should mention in which year of the 15 August to 17 August sample collection were.

4.      Figure number for Synteny analyses of GPX genes in N. sibirica, A. thaliana, and V. vinifera chromosomes was wrongly written as Figure 2. It should be written as Figure 3.

5.      In Figure 4 the figure quality should be improved.

6.       Discussion should be improved by incorporating a greater number of recent articles.

7.       Authors were advice to check whether all references were written as per the journal format or not.

Author Response

Point 1: The introduction is too general, it should be improved.

Response 1: Thank you for the comment. We rephrased the introduction part in a more logical way in the revised manuscript.

Point 2: Authors citation of the plant should be given.

Response 2: Thank you for the comment. We updated citation of the plant in the revised manuscript.

Point 3: In materials and methods: Authors should mention in which year of the 15 August to 17 August sample collection were.

Response 3: Thank you for the comment. This is our oversight, we have added the year of collecting sample in the revised manuscript.

Line 111: The dates of sample collection were 15 August to 17 August, 2022.

Point 4: Figure number for Synteny analyses of GPX genes in N. sibirica, A. thaliana, and V. vinifera chromosomes was wrongly written as Figure 2. It should be written as Figure 3.

Response 4: Thank you for the comment. This is a silly mistake, we have changed the figure number and carefully checked others in the revised manuscript.

Point 5: In Figure 4 the figure quality should be improved.

Response 5: Thank you for the comment. We improved the quality of Figure 4 to make it more accurate.

Point 6: Discussion should be improved by incorporating a greater number of recent articles.

Response 6: Thank you for the comment. We cited several recent articles to improve the discussion part in the revised manuscript.

Point 7: Authors were advice to check whether all references were written as per the journal format or not.

Response 7: Thank you for the comment. We checked all references to make sure all of them were written as per the journal format.

Reviewer 4 Report

1. The manuscript is very well represented.

Some minor changes needed are

2. N sibirica mentioned in line no.13-14 should be included in line no.15

3. Comma (,) in line no.248 should be replaced with a full stop(.), and a separate sentence should be made.

Author Response

Point 1: The manuscript is very well represented. Some minor changes needed are

Response 1: Thank you for the comment. We have checked the spelling and grammar very carefully and corrected some mistakes in the revised manuscript.

Point 2: N. sibirica mentioned in line no.13-14 should be included in line no.15

Response 2: Thank you for the comment. We changed the expression in line no. 13-15 to make the sentence more fluent.

Line 13-15: However,the genome-wide identification of the GPX gene family and its responses to environmental stresses, especially salt stress, in Nitraria sibirica, which is a shrub that can survive in saline environments, has not yet been reported.

Point 3: Comma (,) in line no.248 should be replaced with a full stop(.), and a separate sentence should be made.

Response 3: Thank you for the comment. We have replaced comma with a full stop in line no.248 and divided this sentense into two.

Reviewer 5 Report

The authors of the manuscript titled “The Glutathione Peroxidase Gene Family in Nitraria sibirica: Genome-Wide Identification, Classification, and Gene Expression Analysis under Stress Conditions” The current report contains the executed an analysis to recognize GPX gene family in N. sibirica, and revealed the role of NsGPX in salt resistance physiology.

General comments

Overall, the study is well-designed and presented in a good way.

Abstract

The authors elaborated the abstract in a good way. However, some points need to be addressed to improve this section.

Kindly, add the summary of the current study in the last paragraph of the abstract and the importance of the current investigation in plant sciences.

Introduction

This section is also well written but the authors are requested to study the manuscript carefully and italicize genes and scientific names.

Materials and Methods

The authors are requested to add a reference for the “Plant Materials and Abiotic Stress Treatment” section

Results

The authors are requested to provide gel pictures of the currently studied genes.

Discussion

These sections are well written.

Summary

The summary section of the manuscript should include a summary of the results and discussion, followed by the author’s perspective on how this study will contribute to the current scientific knowledge.  Furthermore, the authors could elaborate on the expected amplification of their work moving forward and explain the relevance of conducting this research in the first place. Providing such insights would enhance the impact and overall value of the manuscript. Therefore, the authors are requested to add future perspective of the current study.

Round 2

Reviewer 2 Report

The authors have reviewed the article properly and I have no further questions.

Reviewer 5 Report

The manuscript is improved and I accept it in its present form.